# A label's a label, no matter the dog: Evaluating the generalizability of the removal of breed labels from adoption cards

**Nicole Passmore Cohen**[1]*, **Martin Chodorow**[2], **Sarah-Elizabeth Byosiere**[1]

**1** Thinking Dog Center, Department of Psychology, Hunter College, City University of New York, New York, New York, United States of America, **2** Department of Psychology, Hunter College and the Graduate Center, City University of New York, New York, New York, United States of America

\* nicole.passmore40@myhunter.cuny.edu

**Data Availability Statement:** All relevant data are within the manuscript and its Supporting Information files.

## Abstract

A common barrier to entry for New York City (NYC) dog adopters trying to rent apartments is the breed label the animal shelter assigned to their dog, despite the fact the labelling is primarily based on intuition and appearance. Bideawee, a limited admission shelter with three locations in the greater New York area, including one in NYC, phased out breed labels from their adoption cards in December 2017. In this study, we evaluated the generalizability of previous findings, specifically, that the removal of breed labels from adoption cards affected length of stay. Moreover, due to Bideawee's multi-location structure, this study provided a unique opportunity to compare variables across different shelter sites while having shelter administration practices held constant. Data from 16-month time periods before and after breed labels were removed was compared. The median length of stay of a dog at Bideawee decreased by 11.3 days (-37.3%) once breed labels were removed ($Mdn$ = 19.0) compared to when breed labels were in place ($Mdn$ = 30.3). A Mann Whitney test indicated that this difference was statistically significant ($U(N_{\text{no breed labels}} = 1259, N_{\text{breed labels}} = 987) = 386309.5$, $z = -15.41$, $p < .001$). Dogs with a "green" behavior assessments (on a scale of green, blue, yellow, red) were almost four and a half times more likely to be adopted faster than "red" dogs (HR: 4.495, 95% CI 2.755–7.335, $p < .001$) before breed labels were removed, but only two times as likely to be adopted faster afterwards (HR: 2.220, 95% CI 1.514–3.254, $p < .001$). The return rate stayed constant across the two time periods at 6%. These findings provide new insights on dog adoptions in the NYC area and suggest that the removal of breed labels will help all dogs get adopted from animal shelters.

## Introduction

While approximately 1.6 million dogs are adopted from animal shelters in the US annually, it is estimated that 3.3 million dogs enter shelters each year. This means that for every dog leaving the shelter system, two dogs are entering. Of those 3.3 million dogs, approximately 20% will be euthanized and 20% will be reclaimed by their owner. The remaining 60%, comprising

**Funding:** The author(s) received no specific funding for this work.

**Competing interests:** The authors have declared that no competing interests exist.

of around 2 million dogs, are available for adoption [1]. For this reason, it is essential to evaluate and identify actions that shelters nationwide can take that increase the percentage of dogs acquired at shelters.

To determine why some dogs get adopted more quickly, and therefore are not at risk of euthanasia or a long stay in a shelter environment, there has been an increase in animal shelter research that examines factors associated with increased adoption success and reduced intake rates [2, 3]. Researchers are taking a critical look at the US shelter system, with scientific publications on animal sheltering increasing from five to ten publications per year in the 1970s to over 50 per year in the present day [3]. To help improve the adoptability of these animals, the investigations have evaluated questions like what kinds of dogs are available for adoption at shelters, where do they come from, and what characteristics and temperaments do they have.

Attempts to capture the characteristics and patterns associated with dogs in shelters at a nationwide level have been largely unsuccessful [4, 5]. A search for statewide shelter numbers yielded similar results; trends are difficult to obtain and are mostly unavailable [3]. As a result, most animal shelter research is conducted locally, in one region, city, or town [6]. Animal shelter research related to population demographics and variables driving adoption success in the US has been conducted locally in New York State [7], Florida [2, 8], Arizona [9], and California [10]. Surprisingly, there are very few studies that examine the population demographics driving adoption success at shelters in major cities like Los Angeles, Chicago, and New York City.

The demographics of local animal shelter populations can be studied to determine which factors contribute to dogs having shorter shelter lengths of stay (time to adoption) or increased live release rates (percentage of adoptions versus euthanasia or natural death). These factors vary widely, even at the local level, for a variety of reasons [3, 7, 9, 10]. Some shelters are limited intake shelters, while others are open admission shelters. Limited intake shelters generally only euthanize animals for critical illness or extreme behavior issues, while open admission facilities euthanize for critical illness, behavior issues and space limitations. However, it is important to note that limited intake shelters sometimes screen a dog's health and temperament before admitting the dog into the shelter, while open admission facilities are required, sometimes by law, to take in animals regardless of temperament, available space, or other factors. As a result, dogs available for adoption at limited intake shelters likely differ physically and behaviorally from dogs available for adoption in open admission shelters. Because of this, as well as the variation in shelter resources across organizations, it is not surprising that different adoption patterns may occur across shelters both locally and nationwide [7].

Regardless of the differences and limitations in evaluating systematic similarities across shelters, it is still possible, and extremely valuable, to conduct research at active animal shelters. At present, phenotypic information like coat color, age, size, and gender continues to be the focus of many local shelter research studies as the outward physical appearance of dogs has been found to correlate with adoption success [2]. This relationship has led researchers to examine the process and outcome of breed labelling dogs. In shelters where the dogs' backgrounds are largely unknown, breed labelling is often determined by staff intuition, prior experiences, or the dog's physical appearance. As a result, how dogs are labelled is inconsistent and can vary by shelter quite drastically [4], especially given that dog professionals do not reliably identify breeds by sight [11].

Despite the lack of structure, science, and agreement in the process of breed labelling, breed labels can still impact a shelter dog's outcome. A study of over 20,000 dogs from an animal shelter in Tucson, Arizona found that the dogs assigned breed labels associated with a stigmatized breed like "Pitbull" had a live release rate of only 80.5%, compared to the live release rate of 91.7% for dogs assigned breed labels that were not considered stigmatized [9]. These

findings have broadly generalized to a second population at a shelter in Orange County, Florida. When breed labels were used, only 52% of Pitbull-type dogs were adopted. When these breed labels were removed, 64% of Pitbull-type dogs were adopted, and the rate of euthanasia for Pitbull-type dogs decreased by a corresponding 12%. These results did not exclusively help improve the adoption rates of Pitbull-type dogs. Adoption rates for all other breed groups also increased following the removal of breed labels [2].

Population demographics and breed labelling are not the only two variables that can impact a dog's length of stay. Another factor researchers have commonly noted as an area for further research is behavior assessments [4, 6, 12]. Behavior assessments aim to predict how suitable a dog will be as a companion, usually by putting the dog through a battery of tests [13]. These assessments aim to simulate the possible challenges a dog might encounter in their future home to inform possible adopters about the dog's behavior. However, such tests are simply approximations of real-world situations, and dogs that are deemed dangerous from a behavior assessment are more likely to be euthanized [14]. While there are standardized behavior assessments, such as the SAFER and BARK assessments [13], which assessment a shelter uses (or even creates themselves) is entirely up to the organizations. Multiple studies have found the predictive ability of behavior assessments to be poor [13, 14]. It has been found that the average rate of false positives (dogs identified as having behavior issues when they actually did not) from behavior assessments in a shelter is 63.8%, and the average rate of false negatives (dogs identified as not having behavior issues when they actually did) is 8.5% [13]. These findings put into question whether behavior assessments truly need to be explored further in future animal shelter research.

A second factor that might impact a dog's length of stay is a shelter's physical location. If a shelter is based in a more suburban or rural area, a high-energy, loud dog may not be as problematic as it could be in an urban setting with small spaces and close neighbors. If the shelter is located near dog-friendly parks and trails, though, people residing in the area may be more open to adopting larger, more active dogs [15]. Socioeconomic and cultural factors of the area where the shelter is based also can impact length of stay and live release [6–8, 15]. In addition, three hundred cities, towns and regions across the US have breed-specific legislation that may drive these breed-labeled dogs out of their new families [4]. While legislation varies from place to place, all limit the acquisition of dogs in some manner, whether it be through a size or weight restriction, by breed label, or from a restriction generated by other phenotypes [4]. Even when dog legislation is not in place city-wide, landlords or management groups can still enact dog restrictions on a residence-by-residence basis. For instance, a study by the ASPCA found that among people living in rental housing, housing issues were the top reason for rehoming a pet [16].

A substantial barrier for pet owners looking to rent or buy in New York City (NYC) is breed labelling in shelters. Landlord pet-related restrictions and dog rehoming are common occurrences in NYC, where the unique combination of limited dog-friendly housing and an abundance of dogs results in many dog owners being put in difficult situations. Approximately 500,000 dogs reside in NYC [17], but a study conducted by StreetEasy, one of the most utilized housing search websites in the NYC area, found that the neighborhood with the greatest share of dog-friendly rental buildings, Battery Park City, only had 63% of rentals open to dogs. This means that there is no neighborhood in NYC where at least two-thirds of the rentals are dog friendly [18]. A survey of pet owners who were relinquishing their large dogs to open admission shelters in NYC found that 32% of the NYC owners had reservations about adopting the dog in the first place due to housing concerns [19]. In addition, the NYC Housing Authority (NYCHA) limits public housing residents to one cat or dog per household. Dogs must weigh less than 25 pounds and must not be a full- or mixed-breed Doberman Pinscher, Pitbull, or

Rottweiler [20]. Beyond public housing, landlords in NYC can request to review the paper-work of a dog adopted from a shelter and reject the dog based on the breed label assigned. The owner then must choose between returning the dog and finding a different apartment, a stress-ful and difficult situation for all involved.

Bideawee, a limited intake, not-for-profit shelter with three locations in the greater New York area, including one in NYC, phased out the use of breed labels on their adoption cards between December 2017 and January 2018. Bideawee's multi-location structure allows for a unique opportunity to compare the characteristics of the dogs at the different shelter sites while having shelter administration and practices held constant. While shelters typically differ in terms of intake, personnel, and policies [21], all three Bideawee locations have the same overarching management and structure. The organization's dog adoption data have never been examined before and, to our knowledge, no study has been conducted on a limited intake, multiple-location animal shelter in New York City. An analysis of their data will both evaluate the impact of the shelter's decision to remove breed labels and contribute new infor-mation to the repository of US local animal shelter studies. In this study, we aimed to review and compare the population demographics of Bideawee's dog adoption data for two 16-month time periods, one when breed labels were being used and one when no breed labels were being used, to determine if length of stay of dogs at Bideawee changed with the removal of breed labels. Moreover, we aimed to analyze the impact that additional factors like phenotypic varia-tions, results of behavior assessments and location of shelter have on length of stay, both before and after breed labels were removed.

## Materials and methods

### Description of data set

All data were collected from Bideawee, a not-for-profit, limited intake shelter for cats and dogs. The organization has three limited intake facilities in New York State: New York City, Westhampton, and Wantagh. Most adoptions are done at the New York City and Westhamp-ton locations, while Wantagh is primarily an intake and administrative facility. Records were obtained from PetPoint (Oakville, ON, CAN), a platform that shelters nationwide use (at a local level) to track the intake and outcomes of animals that come through their organizations.

Data from dogs in two 16-month periods were compared, the "Breed Labels Used" group in which almost all dogs adopted had breed labels on their adoption cards (February 1, 2016 to June 30, 2017) and the "Breed Labels Not Used" group in which almost all dogs had no breed label on their adoption cards (February 1, 2018 to June 30, 2019). Comparing the two corre-sponding time periods allowed for measurement of the impact of the removal of breed labels while limiting the effect(s) of seasonality. A total of 2,508 records were reviewed and 2,246 rec-ords were used in the final analyses (Fig 1). The final data set consisted of 987 dogs in the Breed Labels Used group and 1,259 dogs in the Breed Labels Not Used group (see S1 Dataset for final file used). Note that if a dog was returned to the shelter multiple times within the 16-month time periods evaluated, only the dog's final stay in the shelter was included within the analysis. Dogs that were still in the shelters at the end dates of the two time periods were not included in the analyses. Specific variables for each dog were collected and analyzed, including how the dog entered the shelter (reason), condition at intake, sex, size, age, coat color, behavior code, type of adoption, and place of adoption.

### Place of origin

Bideawee's shelter population consists of dogs brought in by rescue groups both inside and outside the US. Dogs coming into Bideawee could be categorized into one of five regions of

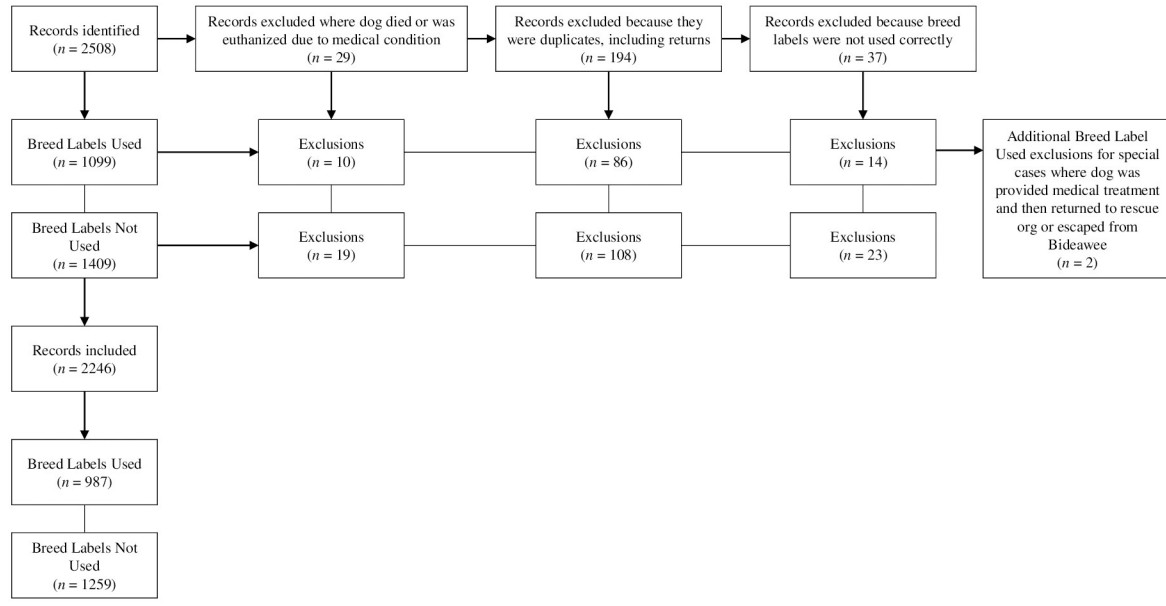

**Fig 1. Data review process.**

origin: New York City, Tri-State Area, the South, California, and Outside the Continental US (Table 1). Dogs whose point of origin was New York City consisted of owner surrenders, owner returns, and strays brought to the Manhattan Bideawee location, as well as dogs taken in from NYC Animal Care & Control. Dogs whose point of origin was the Tri-State Area included owner surrenders, owner returns, and strays brought to the Westhampton and Wantagh locations, as well as dogs taken in from municipal shelters in New York state, New Jersey, and Connecticut. Dogs from the South were those brought in from rescue organizations in Alabama, Georgia, Tennessee, Texas, and Florida. Dogs from California included those brought in from rescue organizations in Los Angeles and San Francisco. It should be noted that Bideawee only began bringing dogs in from California following the increase in wildfires in the state in 2018. As a result, no dogs from California are in the Breed Labels Used group. Finally, dogs could come from abroad, outside of the Continental US, brought in by rescue organizations in the Bahamas, Puerto Rico, Antigua and the US Virgin Islands.

## Reason for entering the shelter

A dog could enter the shelter for three reasons: rescue, owner surrender, or owner return (Table 1). Rescue dogs are either brought in to Bideawee from a rescue organization or, in the case of strays, a concerned citizen. Owner surrenders are dogs not originally adopted from Bideawee that were brought to Bideawee by an owner to give up (note that owner surrenders are charged a $250 processing fee). Owner returns are dogs that were adopted from Bideawee that were returned to the shelter by their adopted owner (there is no fee for returning a dog adopted from the shelter).

## Size, age, and coat color

There were no formal guidelines to assigning size as staff measured this variable in real-time based on intuition and experience. However, puppies were assigned sizes based on the size

**Table 1. Dog population demographics when breed labels were used versus when breed labels were removed.**

| Variables | Breed Labels Used February 2016—June 2017 (N = 987) | | Breed Labels Not Used February 2018—June 2019 (N = 1,259) | |
|---|---|---|---|---|
| | Count | Percentage | Count | Percentage |
| **Place of Origin** | | | | |
| New York City | 70 | 7% | 81 | 6% |
| Tri-State Area | 30 | 3% | 155 | 12% |
| The South | 783 | 79% | 724 | 58% |
| California | 0 | 0% | 49 | 4% |
| Outside Continental US | 104 | 11% | 250 | 20% |
| **Reason** | | | | |
| Rescued | 893 | 91% | 1149 | 91% |
| Owner Surrender | 33 | 3% | 31 | 3% |
| Owner Return | 61 | 6% | 79 | 6% |
| **Intake Condition** | | | | |
| Healthy | 890 | 90% | 1247 | 99% |
| Unhealthy | 97 | 10% | 12 | 1% |
| **Sex** | | | | |
| Male | 469 | 48% | 620 | 49% |
| Female | 518 | 52% | 639 | 51% |
| **Size** | | | | |
| Small | 621 | 63% | 480 | 38% |
| Medium | 256 | 26% | 446 | 35% |
| Large | 110 | 11% | 333 | 27% |
| **Age** | | | | |
| Puppy (< 6 months) | 598 | 61% | 692 | 54% |
| Juvenile (6–12 months) | 47 | 5% | 110 | 9% |
| Young Adult (12–36 months) | 238 | 24% | 310 | 25% |
| Adult (36–96 months) | 93 | 9% | 133 | 11% |
| Senior (> 96 months) | 11 | 1% | 14 | 1% |
| **Coat Color** | | | | |
| Black | 335 | 34% | 408 | 32% |
| Blonde | 31 | 3% | 25 | 2% |
| Brindle | 42 | 4% | 52 | 4% |
| Brown | 298 | 30% | 249 | 20% |
| Grey | 20 | 2% | 23 | 2% |
| Red | 9 | 1% | 38 | 3% |
| Tan | 145 | 15% | 273 | 22% |
| White | 107 | 11% | 191 | 15% |
| **Behavior Color** | | | | |
| No Color | 90 | 9% | 101 | 8% |
| Green | 666 | 68% | 495 | 39% |
| Blue | 19 | 2% | 366 | 29% |
| Yellow | 187 | 19% | 263 | 21% |
| Red | 22 | 2% | 33 | 3% |
| Staff only | 3 | 0% | 1 | 0% |
| **Type of Adoption** | | | | |
| Mobile Adoption | 95 | 10% | 167 | 13% |
| On Site Adoption | 892 | 90% | 1092 | 87% |
| **Site of Adoption** | | | | |

*(Continued)*

**Table 1.** (Continued)

| Variables | Breed Labels Used February 2016—June 2017 (N = 987) | | Breed Labels Not Used February 2018—June 2019 (N = 1,259) | |
|---|---|---|---|---|
| | Count | Percentage | Count | Percentage |
| **Place of Origin** | | | | |
| Manhattan | 666 | 67% | 833 | 66% |
| Westhampton | 304 | 31% | 405 | 32% |
| Wantagh | 17 | 2% | 21 | 2% |

they were expected to be when fully grown. Similarly, dog age, measured in months, was also estimated at intake. Age was then further categorized in five, mutually exclusive, categories (puppy, juvenile, young adult, adult, senior) as proposed by Patronek & Crowe [9]. Coat colors were also further categorized as many possible coat combinations were possible. Eight possible categories were created based on the most common primary coat color (black, blonde, brindle, brown, grey, red, tan, and white).

## Behavior color

Bideawee performs a behavior assessment on their dogs at intake that results in each dog being assigned one of five colors. Dogs can be categorized as green, blue, yellow, red, or white (Table 1). A sticker indicating the behavior assessment color was attached to each dog's adoption card to quickly communicate behavior information to volunteers. However, since the color assigned was noted on the adoption card, potential adopters were able to view this information and may have factored it into their decisions. Green and blue stickers indicated the dog had no restrictions and could be handled by all volunteers. Blue dogs were in between green and yellow dogs and used to be called "mellow yellow," to the point where in the system some dogs initially had their behavior color assigned as "mellow yellow." This was formally switched to blue in 2018 (there is no clear point of transition in the data). Any dog labelled "mellow yellow" had their code changed to blue for this analysis. A true yellow dog (not a "mellow yellow") was deemed to be slightly mouthy, shy, or jumpy, and a volunteer needed additional training (at least 40 hours working with green and blue dogs) before being able to interact with yellow dogs. Red dogs were dogs that demonstrated possessiveness of their toys or food, had a bite history, were excessively jumpy, or that pulled extremely hard on a leash when walked. These dogs could be reactive and aggressive to both humans and animals. As a result, volunteers needed at least 80 hours of experience (40 hours with green and blue dogs, 40 hours with yellow dogs) before being trained to handle red dogs. White color-coded dogs were deemed as staff only, for either behavioral or medical purposes. Note that some dogs never received behavior assessments because they were adopted too quickly or because staff were not available to evaluate the dog. These dogs were coded as "no color assigned" and included in the dataset. This study does not aim to examine the behavior assessment mechanism beyond the assigned color code due to the previously discussed questions around validity and accuracy of behavior assessments. Therefore, all designated behavior color codes will be taken at face value.

## Type and site of adoption

The type of adoption and site of adoption were also evaluated. As Bideawee holds mobile adoption events across the five boroughs of NYC, a dog could be adopted on-site at the shelter or at a mobile adoption event. Moreover, given Bideawee's unique multi-location structure,

on-site adoptions were possible at all three locations, in Manhattan, Westhampton or Wantagh.

## Length of stay

Finally, each dog's length of stay was calculated in days by subtracting the adoption date from the date of intake to determine the total number of days the dog was at Bideawee. Further changes to a dog's length of stay were made based on a dog's holding history, which noted when a dog was on hold (i.e. not available for adoption) for the following reasons: bite quarantine, courtesy hold (meaning the dog was adopted but was staying at the shelter for a day or two longer before going to its new home), media hold (Bideawee uses some of its dogs for photoshoots and TV interviews), medical quarantine, and transfer pending (the dog was being moved to a different Bideawee location). Each dog's number of days on hold were subtracted from their initial length of stay to get a final length of stay that truly captured the time the dog was available for adoption at the shelter.

## Statistical analyses

Data sets were created using Microsoft Excel and data were analyzed with IBM *SPSS* 26. The first step in the analyses was to see if there was a statistically difference average and median lengths of stay between the Breed Labels Used and Breed Labels Not Used group. If there was significance, the impact of other variables on length of stay would be evaluated to isolate the specific impact of breed label use. Non-parametric Mann Whitney tests were conducted to compare the average and median lengths of stay between the two groups, as length of stay is not normally distributed (its distribution skews left) and all of its values are positive (it cannot be less than one). Due to the use of multiple comparisons a Bonferroni correction was used to reduce the risk of Type I error. For comparisons of the length of stay differences between Breed Labels Used and Breed Labels Not Used at the three locations, this correction resulted in a revised alpha level of .0167 to obtain statistical significance.

To evaluate if any variables other than breed labelling impacted length of stay a Cox regression model, a type of survival analysis, was deemed the best fit, as the model could account for non-normal independent variables. Survival analyses examine how much time it takes for an event to occur [22]. In this case, we are examining the time to the adoption for a dog. Cox regression looks at the strength of association between the time to event occurrence and the covariates [22]. In this case, the covariates are breed label use, size, coat color, behavior assessment color, sex, age group, health status, place of origin, mobile adoption (yes/no), and site of adoption. The outcome of a Cox regression model is a hazard ratio (HR) for each covariate. The HR represents the odds of adoption at any time in the model relative to the reference group. For example, for the covariate sex, if female is the reference group and male has an HR of 1.5 (not an actual result), then a male dog is 1.5 times as likely to be adopted at any time in the model relative to a female dog. This suggests that male dogs overall reach their time of adoption faster than female dogs, and therefore overall, male dogs have a shorter length of stay. A statistically significant HR in sex or any other categories could imply that it is several variables, not just breed label use, that impact length of stay.

A key assumption for a Cox regression model is that each covariate in the model meets the test of proportional hazards. If a covariate fails the test of proportional hazards, it can invalidate the model's results entirely. This test checks that the ratio of the hazard functions for two individuals (dogs) in different covariate subgroups does not vary with time. The test of proportional hazards was run in R using cox.zph function [22, 23] and tested all covariates that could be included in the model. Breed label use ($\chi^2$ = 119.34, p < .0001), condition at intake ($\chi^2$ =

8.48, p < .0001) and place of origin ($\chi^2$ = 43.38, $\chi^2$ = 169.92, $\chi^2$ = 11.43 for the different locations; p < .0001 in all cases) were found to be in violation of the test of proportional hazards. As a result, two separate Cox regression models were run in SPSS for the Breed Labels Used and Breed Labels Not Used groups. Condition at intake was removed from the model, as most dogs (90% or more) were healthy in the dataset, and place of origin was designated as a stratifying variable in both models instead of a covariate. Due to the use of multiple comparisons a Bonferroni correction was used in both the Breed Labels Used and Breed Labels Not Used models to reduce the risk of Type I error. This correction resulted in a revised alpha level of .001 to obtain statistical significance.

## Results

The overall median length of stay of a dog was observed to be 11.3 days less (-37.8%) for the Breed Labels Not Used group. The overall mean length of stay was observed to be 8.3 days less (-23.8%). All median and mean lengths of stay by location between the two groups differed by similar quantities, although only the Manhattan and Westhampton differences were statistically significant (Table 2). The Wantagh differences in median and mean length of stay followed the same pattern as Manhattan and Westhampton but had too small a sample size for statistical significance. In all locations and in aggregate, dogs spent less time in the shelter when breed labels were not used.

The Cox regression model aimed to identify any other variables outside of breed label use that could impact length of stay. The model found that size, coat color, gender, age, type of adoption and place of adoption did not have a statistically significant impact on length of stay (Table 3). The only covariate that had a statistically significant effect on a dog's length of stay, in both the Breed Labels Used and Breed Labels Not Used time periods, was behavior assessment color. Dogs with a "green" behavior assessment were almost four and a half times as likely to be adopted on any given day as "red" dogs were (HR: 4.495, 95% CI 2.755–7.335, p < .001) when breed labels were used, but only two times as likely when breed labels were not used (HR: 2.220, 95% CI 1.514–3.254, p < .001). The adoption hazards ratio (HR) for "yellow" and "blue" dogs versus "red" dogs was also less in the Breed Labels Not Used period.

## Discussion

The data analysis showed that median length of stay at Bideawee decreased significantly after breed labels were removed compared to when breed labels were in place. Analyses also showed that dogs with "green" behavior assessments had almost four and a half times the rate of adoption as "red" dogs before breed labels were removed, but only two times the rate afterwards. Despite these significant changes, the data further revealed that the return rate stayed constant at 6% across the two time periods. While correlational, these results further support previous findings that adoption rates for Pitbull-type dogs and other breed groups increase following

**Table 2. Impact of breed labels on length of stay by location, by average days and median days.**

| Location | Count | | Mean length of stay | | | Median length of stay | | | Mann Whitney Test | | |
|---|---|---|---|---|---|---|---|---|---|---|---|
| | Breed Labels Used | Breed Labels Not Used | Breed Labels Used | Breed Labels Not Used | Delta | Breed Labels Used | Breed Labels Not Used | Delta | Mann Whitney U | Z | p |
| **All locations** | 987 | 1259 | 34.9 | 26.6 | (8.3) | 30.3 | 19.0 | (11.3) | 386310 | -15.41 | < 0.001 |
| **Manhattan** | 666 | 833 | 32.9 | 23.7 | (9.2) | 30.3 | 18.0 | (12.3) | 170736 | -12.81 | < 0.001 |
| **Westhampton** | 304 | 405 | 39 | 32.4 | (6.6) | 31.9 | 21.0 | (10.9) | 38577.5 | -8.52 | < 0.001 |
| **Wantagh** | 17 | 21 | 38 | 26.7 | (11.3) | 25.1 | 16.1 | (9.0) | 115 | -1.86 | 0.064 |

**Table 3. Impact of breed labels on length of stay by location, in days.**

| Category | Breed Labels Used HR [95% CI] | Breed Labels Not Used HR [95% CI] |
|---|---|---|
| **Size of Dog**[a] | | |
| **Medium** | 0.885 [0.75–1.045] | 1.109 [0.96–1.28] |
| **Large** | 0.931 [0.751–1.154] | 0.877 [0.739–1.042] |
| **Coat Color**[b] | | |
| **Blonde** | 1.129 [0.778–1.638] | 1.069 [0.706–1.618] |
| **Brindle** | 0.681 [0.488–0.949] * | 0.752 [0.56–1.009] |
| **Brown** | 0.822 [0.699–0.968] | 0.986 [0.837–1.161] |
| **Grey** | 1.475 [0.933–2.332] | 1.03 [0.664–1.598] |
| **Red** | 0.882 [0.444–1.752] | 1.161 [0.822–1.64] |
| **Tan** | 1.164 [0.952–1.423] | 1.348 [1.149–1.582] * |
| **White** | 0.880 [0.703–1.1] | 1.051 [0.873–1.266] |
| **Behavior Evaluation Color**[c] | | |
| **Green** | 4.495 [2.755–7.335] * | 2.220 [1.514–3.254] * |
| **No Color Assigned** | 5.669 [3.356–9.574] * | 2.931 [1.922–4.469] * |
| **Blue** | 4.549 [2.33–8.879] * | 1.846 [1.266–2.693] * |
| **Yellow** | 2.909 [1.772–4.774] | 1.713 [1.171–2.507] |
| **Staff Only** | 0.831 [0.179–3.862] | 2.099 [0.282–15.618] |
| **Male Sex** | 0.897 [0.788–1.021] | 0.998 [0.891–1.118] |
| **Age Group**[d] | | |
| Juvenile | 1.357 [0.979–1.881] | 1.049 [0.841–1.309] |
| Young Adult | 0.968 [0.809–1.159] | 0.876 [0.746–1.027] |
| Adult | 0.846 [0.663–1.08] | 0.762 [0.616–0.943] |
| Senior | 0.49 [0.263–0.91] | 0.633 [0.364–1.098] |
| **Onsite adoption** | 1.302 [1.037–1.633] | 1.04 [0.871–1.241] |
| **Bideawee Site for Adoption**[e] | | |
| Westhampton | 0.858 [0.736–1.000] | 0.905 [0.789–1.039] |
| Wantagh | 0.693 [0.416–1.155] | 0.91 [0.585–1.416] |

* $p < .001$; Stratifying variable = Place of Origin.

[a]Small as reference group.

[b]Black as reference group.

[c]Red as reference group.

[d]Puppy as reference group.

[e]Manhattan as reference group.

the removal of breed labels [2]. Moreover, this trend was observed throughout all three shelter locations within Bideawee's unique multi-location structure, suggesting that these findings are not only generalizable to other limited intake shelters but also might extend to shelters that vary dramatically in location type.

Cox regression models showed that the only characteristic of the dogs adopted at Bideawee that significantly impacted their odds of a faster adoption (and therefore a shorter length of stay) was the behavior assessment color assigned to the dog at intake. The covariate coat color did have a statistically significant effect, but the results were inconsistent. Dogs with a brindle coat color were significantly less likely to be adopted relative to black dogs in the Breed Labels Used period, but there was no significant difference in time to adoption in the Breed Labels Not Used period, while those with a tan coat color were significantly more likely to be adopted relative to black dogs in the Breed Labels Not Used period, but there was no significant

difference in time to adoption in the Breed Labels Used period. One possible reason for these inconsistent results is that dogs were grouped based on their primary color only—for example, both all-black dogs and black-and-white spotted dogs were classified as a black coat color if the black-and-white spotted dogs had a primary color record of black. Future studies could parse these color differences further to investigate additional linkages between coat color and breed labelling, especially since there are myths around both variables (e.g., "is 'black dog syndrome' exacerbated by breed labels?").

The behavior assessment findings showed that dogs with a "red" behavior assessment color were the least likely to be adopted quickly in both time periods, but the odds of "red" dogs being adopted relative to their green, yellow and blue companions were higher in the Breed Labels Not Used period versus the Breed Labels Used period. The importance of the behavior assessment color emphasizes the need to evaluate the standardization, reliability and accuracy of both Bideawee's and other shelters' behavior assessments, as these models indicate that the color assigned can significantly impact how long a dog stays in the shelter environment.

The finding that behavior assessment color also significantly impacted length of stay highlights that the trickiness of using observational data to develop causal inference. This finding could indicate that change in breed label use was the primary reason the lengths of stay were different between the Breed Labels Not Used and Breed Labels Used groups. However, it could also indicate there are several variables influencing lengths of stay. A joint model for longitudinal and time-to-event data would be able to look at both breed label use and behavior assessment color in one model even though breed label use violates the proportional hazards assumption [24].

It is also notable that while the average length of stay was shorter when breed labels were removed from kennel cards, the rate of returns remained constant at 6% across both groups. These findings refute a common myth that removing breed labels limits the amount of information a possible adopter has about a dog and, in turn, will result in more returns. This finding provides additional evidence to support the claim that breed labels do not provide adopters with useful information about a possible future pet. As the findings from this study and the study conducted by Gunter, Barber & Wynne [2] suggest, breed labelling can extend the length of stay of a dog at a shelter, despite there being no scientific basis for a dog's breed assignment. Reducing the length of stay of dogs at shelters puts dogs in homes faster and opens space in shelter facilities, allowing for additional dogs to be rescued. Because of this, shelters that have not removed breed labels from their kennel cards should consider doing so in the future.

## Limitations and future directions

Various limitations to this study do exist. This study was conducted in a New York City limited intake shelter in the United States. The results may not be as generalizable to shelters outside of the United States due to different policies, cultures, and animal welfare approaches [4]. Administrative or personnel factors also may have had an impact on length of stay that cannot be quantitatively accounted for in models or statistical tests. In addition, all PetPoint data was entered by staff during work hours in busy shelter environments, and therefore is subject to possible incorrect data entries due to human error. Another limitation to consider is that behavioral assessment colors were taken at face value and were not analyzed for accuracy or true predictive value. Future studies could evaluate the validity and reliability of these behavioral assessments both within Bideawee and across other limited intake shelters.

The many different approaches for analyzing local shelter data could possibly contribute, in part, to why the results and factors that significantly impact time to adoption vary across different local shelter studies. Causal modelling of observational data could be used to provide further structure to data analyses and help animal welfare researchers determine which variables

directly impact the independent variable (e.g., length of stay) and which variables indirectly impact the independent variable [25]. This could help streamline the number of variables animal welfare researchers examine and make the set of variables more uniform across studies. In addition, research into models such as time-to-event models and accelerated failure time models that make length of stay analyses easy to conduct and understand would be beneficial and drive further uniformity among models in future studies.

Finally, additional studies should be conducted on shelters of a similar profile (limited intake New York City shelters) to identify drivers of dog adoptions and the impact of the removal of breed labels beyond the local level. Similar studies should also be conducted in open admission shelters to determine if results hold across both limited intake and open admission shelters. Accruing more data and results on removing breed labels will strengthen the case for shelters to act, and hopefully, increase the number of dogs placed in loving homes each year.

## Supporting information

**S1 Dataset. Dataset used for analyses.**
(XLSX)

## Acknowledgments

The authors would first like to thank the staff at the Bideawee animal shelter, especially Leslie Granger, Dr. Shian Simms, Ray Cushmore and Deanna Murphy. We also wish to thank Dr. Peter Moller, Dr. Stephen Zawistowski, Julie Hecht, and Jen Abrams for guidance and feedback on previous drafts of this manuscript.

## Author Contributions

**Conceptualization:** Nicole Passmore Cohen, Sarah-Elizabeth Byosiere.

**Data curation:** Nicole Passmore Cohen.

**Formal analysis:** Nicole Passmore Cohen, Martin Chodorow, Sarah-Elizabeth Byosiere.

**Investigation:** Nicole Passmore Cohen.

**Methodology:** Nicole Passmore Cohen, Sarah-Elizabeth Byosiere.

**Project administration:** Nicole Passmore Cohen, Sarah-Elizabeth Byosiere.

**Resources:** Nicole Passmore Cohen.

**Supervision:** Sarah-Elizabeth Byosiere.

**Visualization:** Nicole Passmore Cohen, Sarah-Elizabeth Byosiere.

**Writing – original draft:** Nicole Passmore Cohen.

**Writing – review & editing:** Nicole Passmore Cohen, Martin Chodorow, Sarah-Elizabeth Byosiere.

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
