## [Decision Letter · Decision Letter 0]

8 Jul 2020

PONE-D-20-15309

A label’s a label, no matter the dog: evaluating the generalizability of the removal of breed labels from adoption cards

PLOS ONE

Dear Dr. Passmore,

Thank you for submitting your manuscript to PLOS ONE. After careful consideration, we feel that it has merit but does not fully meet PLOS ONE’s publication criteria as it currently stands. Therefore, we invite you to submit a revised version of the manuscript that addresses the points raised during the review process.

In addition to the reviewer comments, which are detailed at the end of this message, I would like to ask you to consider the following:

1) Lines 135 and 142 Since the study has already been completed, please change "aim" to "aimed)

2) The paper will have an international readership, so reference to US-specific concepts that are not widely known internationally need to be accompanied by enough information to make them understandable. This applies to "501(c)(3)" on line 147 and perhaps also to "Good Samaritan" on line 191 (not clear to me if this term is used in the general sense of somebody doing a good deed for an unrelated individual in need, or if it refers to something more specific).

3) The first paragraph of the discussion is currently only understandable to somebody who has already read the previous text *and *retains important concepts such as LOS, LI and Bideawee's unique multi-location structure. Please consider rewriting, perhaps by adding a first paragraph which sums up the main results in a stand-alone way.

We look forward to receiving your revised manuscript.

Kind regards,

I Anna S Olsson, Ph.D.

Academic Editor

PLOS ONE

Journal Requirements:

2. Please upload a new copy of Figure xxxx as the detail is not clear. Please follow the link for more information: https://blogs.plos.org/plos/2019/06/looking-good-tips-for-creating-your-plos-figures-graphics/" https://blogs.plos.org/plos/2019/06/looking-good-tips-for-creating-your-plos-figures-graphics/" https://blogs.plos.org/plos/2019/06/looking-good-tips-for-creating-your-plos-figures-graphics/

Reviewers' comments:

Reviewer's Responses to Questions

**Comments to the Author**

1. Is the manuscript technically sound, and do the data support the conclusions?

Reviewer #1: Yes

Reviewer #2: Yes

2. Has the statistical analysis been performed appropriately and rigorously? 

Reviewer #1: I Don't Know

Reviewer #2: Yes

3. Have the authors made all data underlying the findings in their manuscript fully available?

Reviewer #1: Yes

Reviewer #2: Yes

4. Is the manuscript presented in an intelligible fashion and written in standard English?

Reviewer #1: Yes

Reviewer #2: Yes

5. Review Comments to the Author

Reviewer #1: I found this a very interesting paper that opened-up some new questions around how information attached to shelter dogs can shape how long they stay in the shelter, or their Length Of Stay. I thought the paper was clearly written and a succinct argument and clearly formed argument through out.

I think the only aspect I would liked to have seen a little bit more discussion or, perhaps in the conclusion is discussion of any research, or to pose research questions, that seek to explain some of the cultures around dog breed/colour type that means that the labelling can exacerbate entrenched assumptions about dogs of a certain breed or colour.

Reviewer #2: General comments

Overall, this paper is excellently presented, with clear goals and rationale, a well designed data collection methodology, clearly written, and conclusions that do not over-interpret the results or over-extend their applicability. It addresses an important topic that requires more research, and contributes to the current literature by providing a generalisable analysis (a semi replication attempt) of a previously-published hypothesis, with a large sample size.

My only issue with the paper is that the statistical analysis has some clumsiness to it. Specifically, it's great that you used survival analysis, which is the natural way to analyse the the data set, but you could have analysed the whole data set (i.e. including the breed label covariate) within one survival analysis model, despite the violation of the proportional hazards assumption. Survival analysis is a very mature field of applied statistical analysis, and there are a number of alternative survival analyses that would have been suitable, such as the accelerated failure time models (where you'd have modelled the the probability of being adopted/still being in the shelter by day x). I think this is important to note because you discuss the different analysis options in both the methods and the limitations section (which I like), but I think you short change survival analysis modelling a bit by not explaining that there are models available to handle the violation of the proportional hazards assumption . For example, I think the class of 'longitudinal time-to-event' models that model a longitudinal data pattern (e.g. the dog's behaviour over time at the shelter) and an event (e.g. adoption) jointly would be a perfect way of analysing shelter dog data in a unified approach (e.g. see the review here https://doi.org/10.1146/annurev-statistics-030718-105048).

Because of the large sample size used, the fact that this is not a within-individual study, and the consistent patterns in the descriptive data showing the effect of breed labels, I think the current analysis is fine for publication (and wouldn't recommend a re-analysis) but I would recommend a few lines in the Discussion/Methods highlighting that you could have used a more sophisticated survival analysis model for your data, which would have reduced the number of models from three to one.

Finally, as mentioned in my specific comment below, it might also be a good opportunity to highlight the need for causal modelling tools in estimating the relationships between shelter behaviour, adoption times, breed etc. I'd recommend the accessible article by Rohrer (2018: https://journals.sagepub.com/eprint/mpV35gbHH9zAw8Zn2wh7/full) for an introduction to how observational data can mislead us. Obviously, this is just a personal preference, but I think it would add to the current literature on this subject, because shelter research is often faced with messy data about which we want to make causal claims.

Specific comments

// Consider whether you need to abbreviate terms like 'length of stay', 'open admission', 'limited intake' -- for me, they obscure the prose and are not needed as they don't appear often enough.

// Line 14: parenthesis around the MW-U test results.

// Line 33: "as a whole"?

// Line 158: what did the data review comprise of, and why did more than 1000 records not meet the conditions for inclusion?

// Line 247: I'm not sure what discussing these other analysis options adds to the paper. Binning the length of stay into a binary variable (line 251) is almost certainly a terrible idea, as it will result in significance loss of power. Similarly, that length of stay is not normally distributed is not really a problem with modern day computing resources (and it is only a problem if one analyses data using a Guassian likelihood if the conditional distribution of the data given the model parameters deviates (far) from a Gaussian distribution). Your mention of 'parametric' models on Line 252 is a bit narrow, implying that parametric is synonymous with 'normally distributed' -- there are many likelihood functions suitable for length of stay-type data, as evidenced by the the numerous types of parametric survival analyses.

// Line 249: "cannot be less than zero" -> "cannot be less than one".

// Line 281: Was the violation of the proportional hazards assumption the reason why you conducted the Mann-Whitney test? If so, I think the paper would benefit from stating this more clearly.

// Table 3: Can the 95% confidence intervals be presented as an inclusive range, e.g. 0.8 [0.7, 0.9] rather than 0.8 (0.7 - 0.9)? The latter implies the interval is exclusive, and could be mistaken for saying the numbers are being subtracted.

// Line 321: I think you should be careful in assuming that breed label use was driving the change based off the fact that behaviour was the only significant covariate in the model. Observational data analysis is extremely messy, and there are numerous unintuitive examples where observational data mislead us (see Pearl (2000) 'Causality' for more). It might be a good opportunity to highlight the benefits of such 'causal modelling of observational data' methods for shelter behaviour research, if it's something you are interested in.

// Line 356: This could be a good opportunity to highlight other methods of survival analysis that you could have used, e.g. accelerated failure time models, time-to-event models that do not consider a proportional hazards assumption.

6. PLOS authors have the option to publish the peer review history of their article (what does this mean?). If published, this will include your full peer review and any attached files.

Reviewer #1: No

Reviewer #2: **Yes: **Conor Goold

---

## [Author Response · Author response to Decision Letter 0]

8 Aug 2020

Please see cover letter and "Response to Reviewers" document.

---

## [Editor Report · Decision Letter 1]

12 Aug 2020

A label’s a label, no matter the dog: evaluating the generalizability of the removal of breed labels from adoption cards

PONE-D-20-15309R1

Dear Dr. Passmore,

We’re pleased to inform you that your manuscript has been judged scientifically suitable for publication and will be formally accepted for publication once it meets all outstanding technical requirements.

Kind regards,

I Anna S Olsson, Ph.D.

Academic Editor

PLOS ONE
---

## [Editor Report · Acceptance letter]

13 Aug 2020

PONE-D-20-15309R1 

A label’s a label, no matter the dog: evaluating the generalizability of the removal of breed labels from adoption cards 

Dear Dr. Cohen:

I'm pleased to inform you that your manuscript has been deemed suitable for publication in PLOS ONE. Congratulations! Your manuscript is now with our production department. 

Kind regards, 

on behalf of

Dr. I Anna S Olsson 

Academic Editor

PLOS ONE